# Mitigating Skin Pigmentation Bias in Pulse Oximetry through Personalized Machine Learning Models

Carter D. Ostrowski[1,2] , Hakan B. Karli[1,3], Bige D. Unluturk[1,3,4]

[1]*Institute for Quantitative Health Science and Engineering*

[2]*Dept. of Biosystems Engineering,* [3]*Dept. of Electrical & Computer Engineering, and* [4]*Dept. of Biomedical Engineering*

Michigan State University, East Lansing, MI, USA

*Abstract*— **Pulse oximeters are essential in neonatal care for monitoring blood oxygen saturation, however their accuracy can be affected by skin pigmentation. The discrepancy between arterial oxygen saturation ($SaO_2$) and saturation measured by pulse oximeters ($SpO_2$) is more pronounced for darker skin tones, increasing the risk of occult hypoxemia. This study introduces a personalized machine learning approach aimed at reducing measurement bias by integrating objective, non-invasive skin pigmentation metrics alongside individual physiological parameters. Using the OpenOximetry Repository, several feature sets were constructed to compare the performance of various machine learning models. XGBoost achieved the lowest root mean square error and was selected for further analysis. The model demonstrated improved $SpO_2$ accuracy, resulting in corrected values which are more closely aligned with actual $SaO_2$ values across a range of skin pigmentation levels. These results support the potential of personalized models to improve measurement accuracy and reduce disparities in clinical monitoring.**

## I. INTRODUCTION

Maintaining adequate blood oxygen levels is essential for ensuring sufficient oxygen delivery to vital organs. This is especially critical in infants, as even brief episodes of hypoxemia, abnormally low oxygen in arterial blood, can lead to acute organ damage and long-term complications [1]. This vulnerability arises from oxygen instability due to the immaturity of their physiological systems responsible for oxygen regulation [2]. In infants, low oxygen saturation is linked to a higher risk of death, neurodevelopmental impairment, patent ductus arteriosus (failure of the fetal ductus arteriosus to close), and necrotizing enterocolitis (inflammatory intestinal injury) [3].To minimize these risks, it is important to accurately assess oxygen saturation in neonatal patients [4].

However, commonly used devices to asses oxygenation, such as pulse oximeters, which estimate peripheral oxygen saturation ($SpO_2$) non-invasively from fingertip or skin, can produce inaccurate readings. This may lead to occult hypoxemia, a condition where arterial oxygen saturation ($SaO_2$, the clinical gold standard) is dangerously low despite normal $SpO_2$ readings [5]. Recent studies have shown that these inaccuracies are not uniformly distributed across demographic groups [6], with preterm infants with darker skin pigmentation leading to a higher risk of occult hypoxemia, potentially resulting in unequal medical care [7].

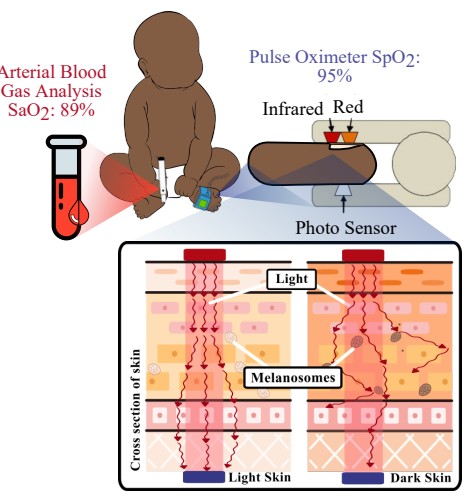

Fig. 1: Pulse oximeter operation relies on light absorption, which differs across individuals due to variations in skin pigmentation.

Compared to arterial blood gas analysis, which requires invasive blood sampling from fragile neonates, pulse oximetry offers a non-invasive approach by estimating arterial oxygen saturation via peripheral measurements [8]. Pulse oximeters function by emitting red and infrared light through the skin and measuring the differential absorption of these wavelengths by oxygenated and deoxygenated hemoglobin. However, in individuals with darker skin pigmentation, elevated melanin content absorbs a greater proportion of incident light, which can attenuate the signal and result in overestimation of oxygen saturation as shown in Fig. 1 [9].

Simple race-based adjustments (e.g. subtracting a fixed "bias" from readings) have proven inadequate in addressing this disparity [10], [11]. Machine learning (ML) approaches have been proposed to address this problem by leveraging extensive datasets such as MIMIC-IV [12] and BOLD [13]. However, these models commonly use race as a surrogate for skin pigmentation, despite the fact that skin tone can vary widely within racial groups and that race itself is a socially defined, subjective classification. In contrast, using direct and objectively measured skin pigmentation provides a more precise and physiologically relevant basis for correcting $SpO_2$ values.

In this paper, we propose a ML framework to mitigate

pulse oximeter inaccuracies through personalized correction of $SpO_2$ measurements. Unlike our prior models that use race as a proxy for skin pigmentation, this study utilizes objective, non-invasive skin pigmentation metrics along with physiological parameters such as age and sex to improve $SpO_2$ estimation accuracy. Leveraging the OpenOximetry dataset [14], we developed an ML model based on XGBoost and evaluated various skin pigmentation metrics that achieve the highest accuracy. We then evaluated our ML model across varying skin tones in the Monk Skin Tone scale (MST) to demonstrate that our model not only performs well on average but also makes an appropriate correction for all skin tones in the Monk Scale.

## II. RELATED WORK

To better understand the origins and implications of measurement bias in pulse oximetry, it is important to examine recent studies that have documented its impact on neonatal populations [7], [9], [15] as well as studies that have proposed potential mitigation strategies [12], [13]. A growing body of literature highlights how racial and skin tone-related disparities in device performance can lead to unequal clinical outcomes [5], [16], [17]. We review key studies that quantify the extent of bias in pulse oximetry [18], [19], analyze regulatory shortcomings [20], [21], and investigate emerging ML approaches developed to address similar limitations in biomedical technologies [22], [23]. It also explores previous attempts to mitigate bias specifically within pulse oximetry, highlighting the limitations of those efforts to inform future solutions [12], [13].

### A. Skin Pigmentation Bias in Pulse Oximetry

Evidence provided in [15] shows that pulse oximeters systematically overestimate oxygen saturation in Black infants and children, increasing the risk of occult hypoxemia in these populations. In the study 12% of Black children with confirmed hypoxemia ($SaO_2$ <88%) were misclassified as having normal oxygen levels ($SpO_2 \geq$ 92%), compared to only 4% of White children, out of the 774 pediatric cardiac patients. Similar disparities were observed in [7] among pediatric COVID-19 cases, where infants born to Black mothers exhibited a higher average overestimation of $SpO_2$ (2.22%) than those born to White mothers (1.41%). These findings indicate the measurement bias that affects racially diverse neonatal and pediatric populations.

As a result, clinically significant discrepancies in oxygen readings among infants with darker skin pigmentation may go undetected within aggregate performance metrics [19]. To build on this, [7] conducted a prospective study analyzing 4,387 matched $SaO_2$–$SpO_2$ measurements from 294 infants born before 32 weeks of gestation and found a significantly greater overestimation of oxygen saturation in Black infants (1.73%) compared to White infants (0.72%). This discrepancy increased to 2.22% within the $SpO_2$ range of 85–100%. Occult hypoxemia occurred more frequently in Black infants (9.2%) than in White infants (7.7%). Unlike population level studies, these results provide patient level

evidence of inconsistency in pulse oximetry performance, raising concerns about the accuracy of day to day clinical monitoring in neonatal intensive care settings.

Nevertheless, the Food and Drug Administration (FDA) permits pulse oximeters for approval with a reported average error of no more than 3%, based on testing cohorts that include 15% of individuals with darker skin pigmentation [20]. In response to growing evidence of the racial disparities, recent regulatory efforts by FDA have aimed to revise the evaluation protocols to ensure more consistent accuracy across diverse skin pigmentations [21].

### B. Bias in Other Biomedical Technologies

Measurement bias in pulse oximetry exemplifies a broader limitation of traditional biomedical tools in capturing individual physiological variability [24]. ML offers a promising solution by identifying subtle, patient specific patterns in physiological signals. For example, in major depressive disorder, conventional EEG-based approaches have failed to produce reliable biomarkers for treatment selection [25]. A meta analysis by [22] examines the use of a ML models as a solution. The models successfully extract clinically relevant features from EEG data to predict antidepressant response, and thus improving diagnostic precision. Similarly, ML has enhanced the accuracy of infrared thermography (IRT) for fever detection, a technology also affected by skin pigmentation and motion artifacts. In [23] it is displayed how a Random Forest model trained on multi-region facial temperature and ambient data significantly reduced core temperature prediction error, from 0.61°C to 0.24°C. This demonstrating how ML based corrections can account for variability in optical sensing systems, like pulse oximetrs.

### C. Correcting Bias in Pulse Oximetry

Due to the dynamic errors in pulse oximetry such as directional inconsistencies and differential inaccuracies, a static correction factor is not an appropriate solution to bias [10], [11]. Hence, ML approaches have been proposed to debias widely used pulse oximeter readings by using patient demographics. An effort by [12] was conducted, where an XGBoost regression model was developed using paired $SaO_2$ and $SpO_2$ data with an attempt to reduce racial bias in pulse oximeter measurements. There was a particular focus on addressing occult hypoxemia in patients with darker skin pigmentation. The model achieved a substantial improvement in predictive accuracy among Black patients, with the $R^2$ increasing from 21.8% to 67.6%. Despite its promising results, the study's broader clinical applicability remains limited. The training dataset underrepresented key demographic groups, raising concerns about generalizability. Moreover, the model relies on features derived from invasive blood analyses, such as hemoglobin concentration, which undermines the primary advantage of pulse oximetry as a non-invasive and easy to use monitoring tool.

To facilitate research in this area, more datasets focused on skin pigmentation bias in pulse oximeters emerged such as BOLD [26], ENCoDE [27], and OpenOximetry [14].

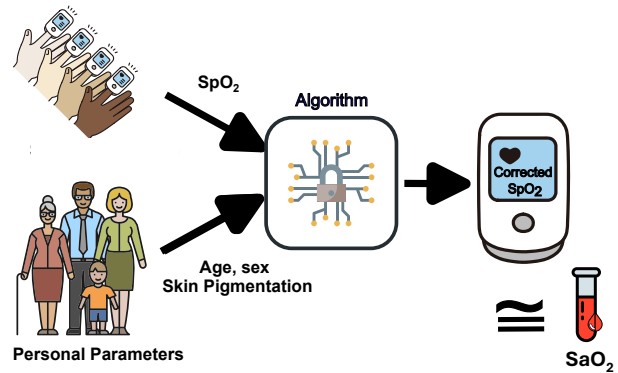

Fig. 2: Personalized pulse oximeter correction to estimate SaO$_2$ better.

Another important study addressed the racial bias in pulse oximetry by developing an XGBoost regressional model trained on the BOLD dataset. The model estimated SaO$_2$ from SpO$_2$ using a combination of interpersonal features, including race as a surrogate for skin pigmentation. The approach significantly improved predictive accuracy, reducing mean squared error from 30.76 to 4.72.

However, the use of race rather than direct measures of skin pigmentation presents a limitation, as it may not adequately capture individual variation in melanin concentration. This reliance could limit the model's generalizability and effectiveness in addressing pigment driven discrepancies in a neonatal care settings.

This work addresses skin pigmentation bias in pulse oximetry through a personalized ML framework. Unlike previous models that use race as a surrogate for skin pigmentation, our approach integrates objective, non-invasive skin pigmentation measurements alongside personal physiological parameters to enhance the accuracy of oxygen saturation estimation. By explicitly accounting for variability in skin pigmentation, the model aims to improve SaO$_2$ prediction reliability across diverse patient populations and to mitigate discrepancies that disproportionately affect individuals with darker skin pigmentation, as visualized in Fig. 2.

## III. METHODOLOGY

We used the OpenOximetry Repository (version 1.0.1) [14], which is a structured and curated database designed to consolidate clinical and laboratory pulse oximeter data. It contains 8,614 paired SpO$_2$-SaO$_2$ measurements from desaturation encounters, where patients' oxygen saturation is varied to stable targets between 70 and 100%, by having them breathe medical air adjusted to different levels of oxygen [14]. Additionally, the dataset includes comprehensive patient demographic data, including both qualitative and quantitative skin pigmentation measurements taken at various anatomical sites.

In total, the data represents 263 controlled desaturation encounters across 100 unique patients, each providing 20–30 samples per session. The patient metadata includes age, sex, and four skin tone scales: Fitzpatrick, MST (Fig. 3(a)) [28], Melanin Index, and the Munsell Color System (Fig. 3(b))

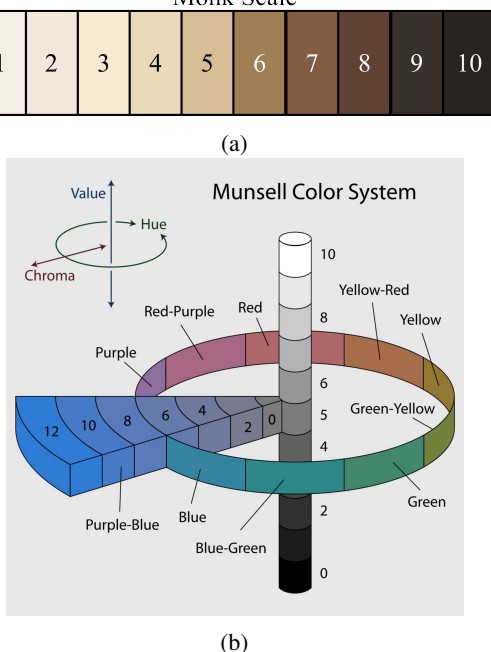

Fig. 3: (a) MST [28] (b) Munsell color system [29].

[29]. Addition device specific measurements like perfusion index (PI) are included. The dataset contains a total of 8,614 data readings from 32 distinct devices; however, not all 32 devices contributed to every data point. Devices 59 and 60 account for over 7,700 data points each, whereas the next most frequently used device appears in only about 1,400 data points. Compared to device 59, the measurements from device 60 have a slightly higher proportion of patients with darker skin pigmentation, helping to reduce class imbalance. For Device 60, the distribution of skin tone groups is illustrated in Fig. 4.

### A. Data Preprocessing and Feature Selection

Duplicate measurements were removed while retaining the sample with the most complete information across key fields. Device 60 was selected for analysis due to its high volume of readings and broad representation across skin tone categories. Specifically, the dataset comprises 7,747 measurements from 97 patients. Of these patients, 57.73% are male and 40.21% are female. The patient age range spans from 17 to 47 years, with the majority of encounters (73.03%) occurring in individuals aged 21–30.

Skin tone data (excluding unknown) from the four measurement scales (Fitzpatrick, MST, Melanin Index, and the Munsell Color System) was included, along with the personal attributes of age and sex. In addition, measurement take only from the fingernail site were used, to ensure consistency.

The categorical variables (sex, Fitzpatrick, MST, and Munsell hue) were encoded for compatibility with machine learning algorithms. RGB values were extracted from the Munsell Color System to provide a standardized numerical color representation for comparison. Outliers in SpO$_2$ readings were identified and removed, and SpO$_2$ values were z transformed to enhance model training performance.

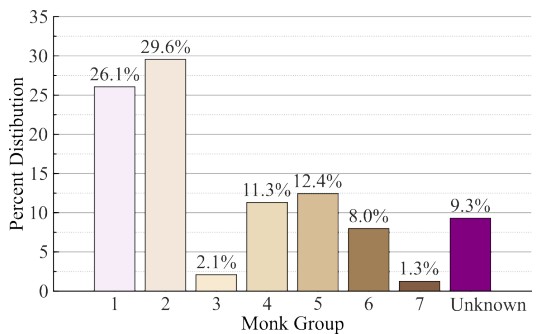

Fig. 4: Distribution of skin pigmentation in the data for Device 60, categorized using MST.

### B. Feature Sets

To assess model performance, we developed seven distinct feature sets, described in Table I. The feature sets comprise of $SpO_2$ data, the different skin pigmentation scales, age, and sex. As the feature set number increases, the resolution of describing skin pigmentation level is increasing. For example, feature set 3 using Fitzpatrick scale has only 7 skin pigmentation categories whereas MST has 10 categories. Also feature sets 3 and 4 use categorical skin pigmentation representation where feature sets 5, 6, and 7 use continuous representation. We also created another version of these sets with PI included in all sets.

TABLE I: Feature sets.

| Set | SpO$_2$ | Age | Sex | Fitz | Monk | M.I. | Muns | RGB |
|---|---|---|---|---|---|---|---|---|
| 1 | ✓ | | | | | | | |
| 2 | ✓ | ✓ | ✓ | | | | | |
| 3 | ✓ | ✓ | ✓ | ✓ | | | | |
| 4 | ✓ | ✓ | ✓ | | ✓ | | | |
| 5 | ✓ | ✓ | ✓ | | | ✓ | | |
| 6 | ✓ | ✓ | ✓ | | | | ✓ | |
| 7 | ✓ | ✓ | ✓ | | | | | ✓ |

To identify the optimal predictive model, we evaluated a variety of machine learning algorithms across the seven feature sets. These included tree-based models (Decision Tree, Random Forest, Gradient Boosting, and XGBoost) which leverage hierarchical structures and ensemble techniques to capture complex, non-linear relationships in the data [30]. We also considered linear regression models (Elastic Net, Lasso Regression, and Ridge Regression) which incorporate regularization to manage multicollinearity and reduce overfitting [31]. Lastly, we explored non-linear and advanced models (K-Nearest Neighbors or KNN, Neural Networks, and Support Vector Regression (SVR)) which offer flexible and adaptive approaches capable of modeling intricate physiological data patterns [32]. For each model on each feature set, the hyperparameters of the were tuned to achieve the lowest root mean squared error (RMSE).

### IV. Results

Model performance was quantified through RMSE across all feature sets to identify the best model. For each model on each feature set, the hyperparameters of the were tuned to achieve the lowest root mean squared error (RMSE). The RMSE, between $SaO_2$ and measured $SpO_2$, of the

dataset before correction is 3.41. Among the ML models we implemented, XGBoost produced the best RMSE in feature set 7, as shown in Table II, reaching an RMSE of 1.03. The hyperparameter of the XGBoost model are 214 estimators, a learning rate of 0.1511, a maximum depth of 7, a subsample of 0.8391, a colsample_bytree of 0.9319, and a gamma of 0.001. Although gradient boosting performed best on feature sets one through six, XGBoost was selected for further analysis due to it producing the best RMSE. This result aligns well with previous pulse oximeter correction studies [12], [13] that also selected XGBoost as best performing model.

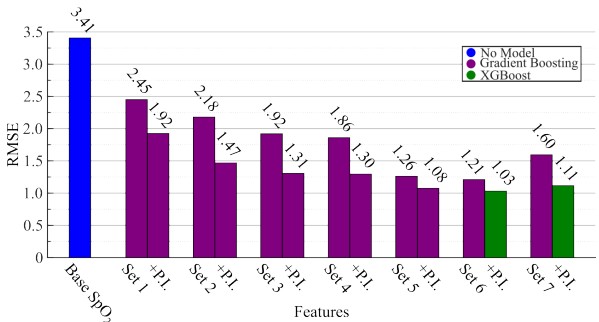

Fig. 5: Best model performance across the differing feature sets with the addition of perfusion index (PI).

TABLE II: RMSE values for ML models across feature sets.

| Model | 1 | 2 | 3 | 4 | 5 | 6 | 7 |
|---|---|---|---|---|---|---|---|
| Decision Tree | 1.996 | 1.856 | 1.702 | 1.704 | 1.520 | 1.501 | 1.548 |
| Elastic Net | 2.283 | 2.279 | 2.214 | 2.182 | 2.188 | 2.154 | 2.070 |
| Gradient Boosting | **1.925** | **1.470** | **1.306** | **1.296** | **1.076** | 1.035 | 1.131 |
| KNN | 2.037 | 1.684 | 1.629 | 1.678 | 1.472 | 1.738 | 1.784 |
| Lasso Regression | 2.283 | 2.279 | 2.214 | 2.182 | 2.188 | 2.154 | 2.070 |
| Linear Regression | 2.283 | 2.279 | 2.214 | 2.182 | 2.188 | 2.154 | 2.070 |
| Neural Network | 2.000 | 2.009 | 1.937 | 1.900 | 1.896 | 1.846 | 2.544 |
| Random Forest | 1.929 | 1.565 | 1.414 | 1.418 | 1.259 | 1.177 | 1.257 |
| Ridge Regression | 2.283 | 2.279 | 2.214 | 2.182 | 2.188 | 2.154 | 2.070 |
| SVR | 1.996 | 1.685 | 1.540 | 1.466 | 1.505 | 1.254 | 1.384 |
| XGBoost | 2.036 | 1.517 | 1.322 | 1.299 | 1.087 | **1.032** | **1.115** |

Since PI showed to consistently improved performance, it was added to each of the feature sets shown in Table II. XGBoost performance with and without PI is shown in Fig. 5.

As the resolution of skin pigmentation representation increased, model accuracy improved. For example, feature sets 3, which used a less detailed skin tone scale, resulted in an RMSE of 1.31, while feature sets 6, with a more detailed representation, achieved an RMSE of 1.03. PI further increased accuracy, contributing more in feature sets with lower resolution.

To address the concern that the correction model may perform well on average but fail to correct errors within specific demographic subgroups, we conducted a detailed subgroup level evaluation. The comparison of measured $SpO_2$ and corrected $SpO_2$ with respect to $SaO_2$ for all datapoints is illustrated in Fig. 6. The data is broken down by each MST group. Since the dataset did not contain data for MST groups 8-9, the results are plotted for MST 1-7.

In Fig. 6 (a) and (b), the overall result for all MST groups is shown. It is observed that the corrected $SpO_2$ aligns better with $SaO_2$. In Fig. 6, panels (c)–(p) provide the detailed

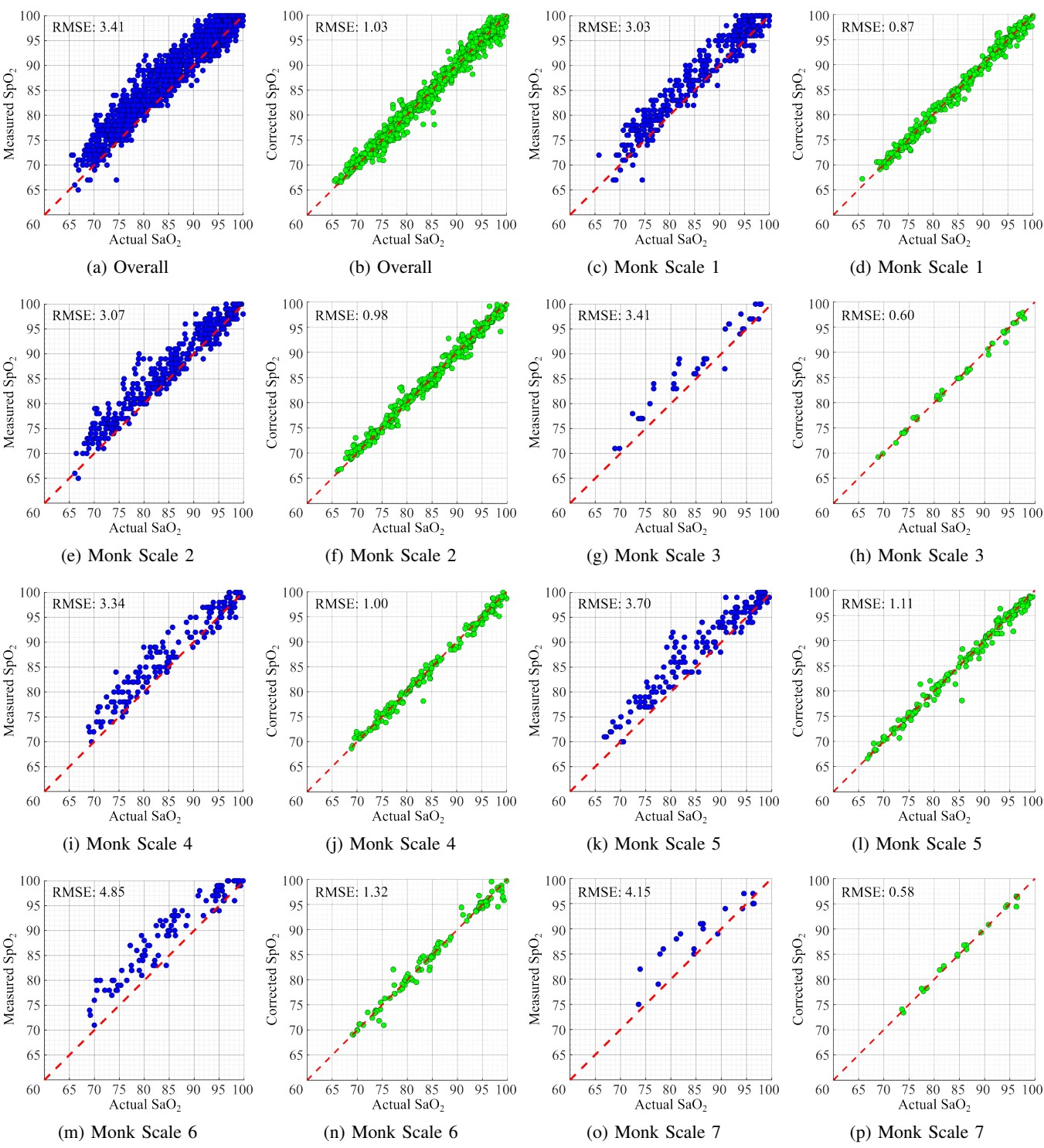

Fig. 6: Comparison between measured SpO$_2$ and corrected SpO$_2$ generated by the XGBoost model with respect to SaO$_2$. (a) and (b) show the overall model performance before and after correction. (c)–(d) through (o)–(p) show the measured SpO$_2$ (left) and corrected SpO$_2$ (right) for subjects in MST scale group 1 through 7, respectively.

breakdown of model performance across individual MST groups, highlighting the differences between the measured and corrected SpO$_2$ values. As skin pigmentation becomes darker, there is a noticeable trend of increasing average overestimation of blood oxygen saturation levels, along with increasing RMSE from 3.03 in Fig. 6(c) to 4.85 in Fig. 6 (m) for measured SpO$_2$.

With the XGBoost corrections, RMSE values stayed around an RMSE of 1.00 across all MST groups, with minor variation likely due to data imbalance. Corrected SpO$_2$ aligned consistently with SaO$_2$ along the 1:1 line, showing improved performance across skin tones without degradation from skin pigmentation. To better illustrate this improvement, Fig. 7 shows Bland-Altman analysis comparing measured

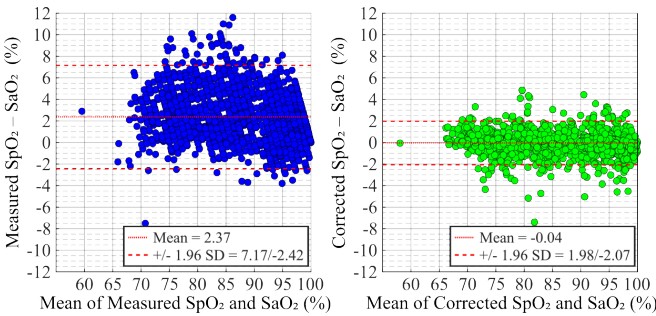

Fig. 7: Bland Altman plots for the measured (left) and XGboost corrected (right) SpO₂ data.

and corrected SpO₂ with SaO₂. The measured values show a mean bias of 2.37 and wide limits (7.17, –2.42), further displaying the overestimation along with the high variability. In contrast, the corrected values have a mean of –0.04 and tighter limits (1.98, –2.07), showing closer alignment with SaO₂. The correction reduces both bias and variability, making corrected SpO₂ values more accurate and thus better aligned with SaO₂. To assess the model's cross device applicability, it was trained on data from another pulse oximeter (Device 59) and its corresponding measured SpO₂ values, as shown in Fig. 8. The model was trained with feature set 7. The corrected values led to some improvement, but error remained relatively high and overestimation persisted. For reference, the measured SpO₂ RMSE for Device 59 was 2.38. These results suggest that device-specific training is needed for more accurate estimations. Just as models must be tailored for individual variation, generalization across devices is limited. Therefore, training should be adapted to each device to ensure consistent performance.

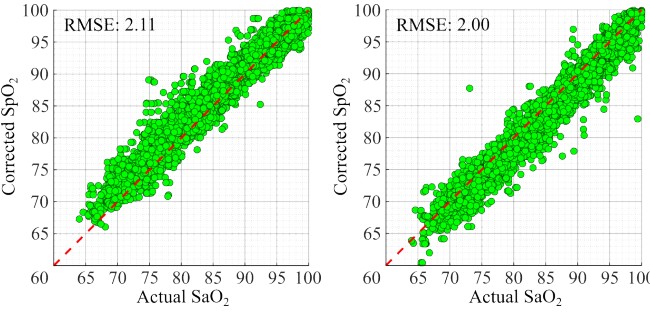

(a) Trained on Device 59, tested on Device 60.
(b) Trained on Device 60, tested on Device 59.

Fig. 8: Evaluating model generalizability across devices.

Another challenge highlighted in the literature is correctly estimating fluctuations in pulse oximeter readings for the same individual over time, due to its disparities present for black patients [11]. To evaluate this, corrected SpO₂ was plotted as a time series, displaying changes in SaO₂, measured SpO₂, and corrected SpO₂ during an encounter. This is shown in Fig. 9 with an example encounter from each MST group. For all MST groups, the model's corrected SpO₂ values more closely follow the SaO₂ values, offering improved estimations across a wide range of samples and oxygen saturation levels. In contrast, measured SpO₂

consistently overestimates oxygen saturation, including cases that might cause occult hypoxia. For example, in Fig. 9 (c) and (d), SaO₂ is around 83%, while the measured SpO₂ is approximately 90%, while the corrected SpO₂ values remain within approximately 1% of the SaO₂. The corrected SpO₂ values help better identify hypoxic conditions that might otherwise be missed. In these cases, relying on the measured SpO₂ may lead to missed detections of occult hypoxia.

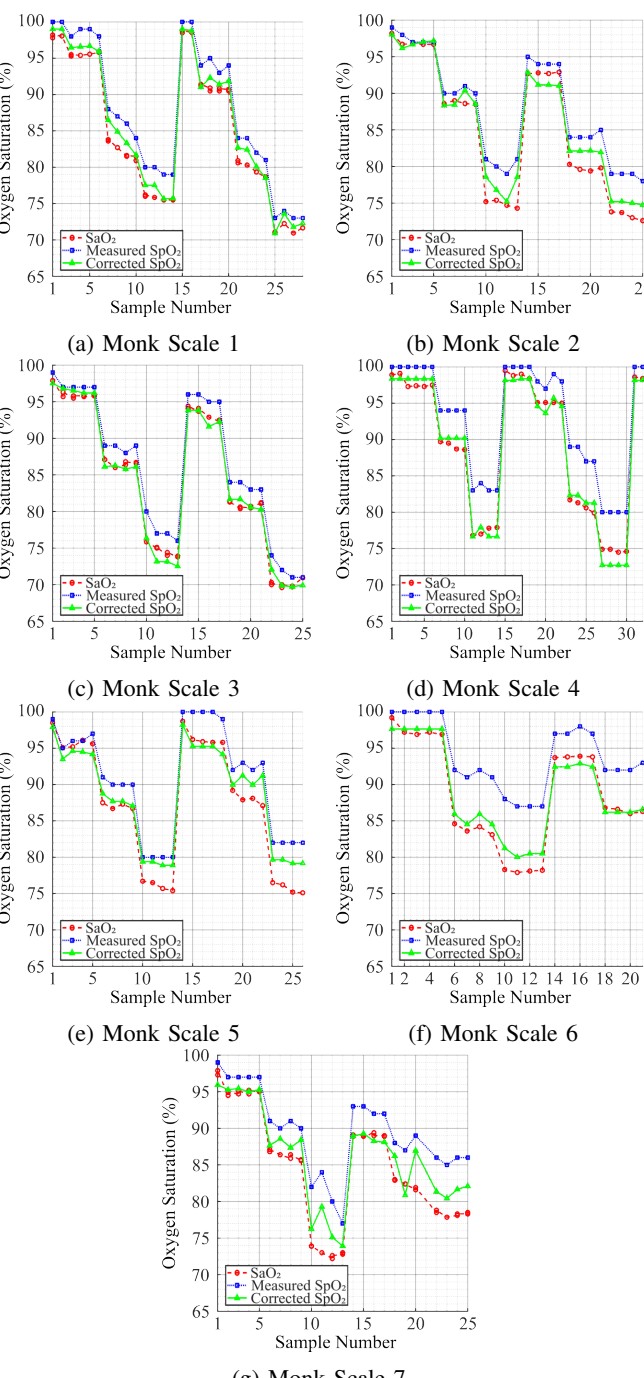

(a) Monk Scale 1
(b) Monk Scale 2
(c) Monk Scale 3
(d) Monk Scale 4
(e) Monk Scale 5
(f) Monk Scale 6
(g) Monk Scale 7

Fig. 9: Temporal variation of SaO₂, measured SpO₂, and corrected SpO₂ from a single individual across sequentially collected samples. Panels (a)–(g) represent data from MST scale Groups 1 to 7.

## V. Conclusion

This study demonstrates that the accuracy of pulse oximetry can be substantially improved by integrating individual physiological parameters and objective skin pigmentation metrics into machine learning based predictive models. The XGBoost model proved to significantly reduce measurement bias across all skin tone groups present and maintained high accuracy, closely following $SaO_2$, outperforming the measured $SpO_2$ estimates. These results display the limitations of traditional generalized approaches and highlight the potential of personalized algorithms to promote equity in medical monitoring technologies.

## VI. Acknowledgments

This material is based upon work supported in part by the National Institutes of Health (NIH) under Grant 1R01HL172293-01 and in part by 1R21EB036329-01.

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
