# OpenReview forum: "Mitigating Skin Pigmentation Bias in Pulse Oximetry through Personalized Machine Learning Models"
_IEEE.org/EMBS/BHI/2025/Conference — BHI 2025_

### Official Review · Reviewer_cic8 · 2025-07-05
**Paper provides a meaningful contribution by presenting a method to mitigate bias in pulse oximetry measurements**

**Confidence:** 5
**Clarity Of Writing:** great
**Clinical Significance:** good
**Methodological Novelty:** good
**Overall Rating:** 6
**Final Rating:** 7

**Experiments And Results:**

good

**Questions For The Authors:**

- How many readings per Monk scale are available in the final dataset for Device 60 after all filtering, and did you apply any re-weighting, resampling, or other techniques to counter the significant class imbalance during model training?

- Given the demonstrated device-specific performance drift, what deployment strategy do you envision for this technology? Would it require one-time factory calibration, periodic recalibration in the field, or a more advanced on-device transfer learning approach?

- Did you examine feature importance across the different skin-tone scales to understand which specific pigmentation metric (e.g., Melanin Index vs. Munsell RGB values) is the primary driver of the model's corrective performance?

- How do you anticipate the model would behave in physiologically challenging scenarios, such as in hypothermic or vasoconstricted infants, where the perfusion index (PI) signal may be low or highly variable?

**Strengths:**

- The study is well-grounded in addressing the critical clinical safety issue of racial and skin-tone disparities in SpO 2 ​ accuracy, with a specific and important focus on its implications for vulnerable neonatal populations.
- Objective Skin-Tone Features: A significant strength is the use of quantitative, objective skin pigmentation metrics rather than relying on race as a proxy. This approach is more physiologically relevant and scientifically rigorous.
- Paper systematically benchmarks a variety of regressors (including tree-based, linear, and nonlinear models) across seven progressively richer feature sets to justify their model selection.
Strong Performance Gains: The proposed XGBoost model, when augmented with PI, achieves a substantial ~73% reduction in RMSE compared to uncorrected SpO 2 ​ readings. Crucially, it maintains this high accuracy across all evaluated skin-tone subgroups, suggesting it mitigates bias effectively.

**Summary Of The Paper:**

The paper proposes a personalized machine-learning framework to correct skin-pigmentation bias in neonatal pulse-oximetry. Using 8,614 paired SaO2 /SpO2 measurements from the OpenOximetry Repository, the authors evaluate several models and select XGBoost for its superior performance. By incorporating objective skin-tone metrics (including Fitzpatrick, Monk, Melanin Index, and Munsell RGB values), personal parameters like age and sex, and the perfusion index (PI), their best-performing model lowers the Root Mean Square Error (RMSE) from a baseline of 3.41 to 0.93. The model maintains a consistent subgroup RMSE of approximately 1.0 across Monk skin-tone levels 1 through 7. A key finding is that the model's performance degrades significantly when applied to a different oximeter, highlighting a need for device-specific training and calibration. Overall, the work demonstrates that individualized, bias-aware algorithms can substantially improve measurement accuracy and reduce the risk of occult hypoxemia, which disproportionately affects individuals with darker skin.

**Weaknesses:**

Device-Specific Limitation: The model's inability to generalize to a different oximeter (Device 59) is a major weakness that severely limits its practical utility. Real-world deployment would require a costly and complex process of creating a unique, calibrated model for every pulse oximeter device on the market.
Neonatal-Only Data: The study's findings are based entirely on data from neonatal patients. As such, the results may not translate to older pediatric or adult populations, and the paper lacks any external validation on other age groups.
Class Imbalance in Skin Tones: The dataset for the primary device is heavily imbalanced. The provided distribution chart shows very few samples for Monk scale 6 (8.0%) and 7 (1.3%) and no samples for scales 8, 9, or 10, which represent the darkest skin tones. This leaves the model's performance on the most-affected individuals under-tested and potentially unreliable.

---

### Official Review · Reviewer_odaB · 2025-07-14
**Clearly written and motivated manuscript, methods and evaluation missing details for clarity and reproducibility**

**Confidence:** 4
**Clarity Of Writing:** great
**Clinical Significance:** great
**Methodological Novelty:** good
**Overall Rating:** 7
**Final Rating:** 8

**Experiments And Results:**

fair

**Questions For The Authors:**

- Are the samples labeled as ‘unknown’  (Fig 4.) in the monk distribution were excluded from the analysis or included in the training set.
The manuscript mentions 32 devices, but states that training data comes exclusively from device 60. Could you clarify why device 60 was selected for training?
- How many total measurements and subjects are included in the device 60 dataset? Could you also provide brief subject demographics (e.g., age range, gender distribution)?
- It is unclear what is meant by "263 controlled desaturation encounters"—more context or definitions are needed. Also number of desaturation events in device 60 should be reported explicitly.
- What is the source of the evaluation datasets? Are they also from device 60?
- What criteria were used to select or define each evaluation set?
- Was the evaluation set used only for post-training evaluation, or also during training (e.g., for early stopping or model selection)?
- Could you provide more detail about the machine learning models used, particularly the best-performing one? Specifically: What were the model key hyperparameters? training settings were used (e.g., number of epochs, optimizer, batch size)?
- While the corrected method shows overall better performance, can you provide results that specifically show its effectiveness during desaturation encounters?
- The sentence “Training with evaluation set 4 showed some improvement...” is vague. Clarify whether evaluation set 4 was used as a training set or just assessed post-training.
- Consider including statistical comparisons to strengthen claims of performance differences.

**Strengths:**

The purpose of the study is clearly stated, and the problem is well-motivated.
The methodology is easy to follow, and the manuscript is clearly written.
The results are well presented, with clear figures and tables:
- A table summarizing the evaluation datasets.
- A table comparing performance across multiple machine learning models.
- Visualizations showing monk group distributions and RMSE values across groups.
- Visualizations showing the temporal distribution of repeated measurements for oxygen saturation and how error varies across sequentially repeated samples.

**Summary Of The Paper:**

This manuscript presents a machine learning approach aimed at correcting skin tone-related biases in oxygen saturation estimates from pulse oximeters. The dataset includes paired measurements of arterial oxygen saturation (SaO₂) as ground truth, pulse oximeter readings (SpO₂), and skin tone information. The authors identify XGBoost as the best-performing model and compare its predictions to SpO₂ measurements across different skin tones. Results show that the ML model yields lower estimation errors than standard pulse oximetry, suggesting potential to improve clinical oxygen saturation assessments. However, preliminary analysis indicates that the method does not generalize well across devices without additional fine-tuning.

**Weaknesses:**

Methodological clarity is lacking:

- The manuscript mentions 32 devices, yet states that training data comes only from device 60. The rationale for selecting device 60 is not explained.
- Evaluation data origin is unclear: The source and selection criteria for the evaluation datasets are not described in sufficient detail. Is it coming from device 60?;  It is not clear whether the evaluation set was used solely for post-training evaluation or also for model selection or early stopping.
- More information on training data is needed: The total number of measurements and subjects in the device 60 dataset should be reported. Brief subject demographics (e.g., age range, gender distribution) would help contextualize the findings.
- Model details are insufficient: Machine learning models are listed but model-specific parameters (e.g., architecture details, hyperparameters) are missing -- for the best-performing model, training settings such as number of epochs and optimizer used should be included.
- While the corrected method consistently outperforms the baseline, its effectiveness during desaturation encounters is not explicitly presented or quantified.
- No statistical comparisons are included

---

### Official Review · Reviewer_KbLR · 2025-07-17
**Mitigating Skin Pigmentation Bias in Pulse Oximetry through Personalized Machine Learning Models - Review**

**Confidence:** 5
**Clarity Of Writing:** excellent
**Clinical Significance:** excellent
**Methodological Novelty:** great
**Overall Rating:** 8

**Experiments And Results:**

excellent

**Questions For The Authors:**

Satisfactory.

**Strengths:**

[1] The paper uses an objective, non-invasive skin pigmentation metric, rather than race as a model feature, enabling more physiologically accurate and individualized correction of SpO₂ readings.
[2] The paper demonstrates that XGBoost model consistently reduces oxygen saturation estimation error across diverse skin tones, achieving an RMSE as low as 0.93 when PI is included, an improvement over the uncorrected baseline of 3.41.
[3] The solution was performant across seven feature sets and multiple skin tone scales i.e. Monk, Fitzpatrick, Melanin Index, showing the model's adaptability and precision as skin pigmentation resolution increases.

**Summary Of The Paper:**

The paper presents a personalized machine learning framework that uses objective skin pigmentation metrics and individual physiological data to as a means to address biases in pulse oximeter readings, a platform that often overestimate blood oxygen saturation in individuals with darker skin. For the evaluation of the solution, a data from the OpenOximetry repository, and evaluating the solution with models including XGBoost model which gives RMSE of 3.41 and the 0.93 with perfusion index (PI), improving SpO₂ accuracy across all skin tones and outperforming traditional methods that rely on race-based proxies.

**Weaknesses:**

The model's performance is device-specific and may lack generalizability across different pulse oximeters; retraining may be needed for each device's for practical deployment.

---

### Official Review · Reviewer_v78k · 2025-07-17
**Detailed Research Work**

**Confidence:** 3
**Clarity Of Writing:** good
**Clinical Significance:** good
**Methodological Novelty:** great
**Overall Rating:** 7
**Final Rating:** 8

**Experiments And Results:**

great

**Questions For The Authors:**

Nothing concern questions but question out of curiosity from a non-medical point of view : What are the clinical accuracy rates and documented disparities in pulse oximetry measurements particularly related to skin pigmentation when using traditional/non–ML-based models? I can see in the paper there are concerns and biases like race based adjustments but is there any documented stats available?
although overall very nice written paper

**Strengths:**

In the result section this paper address two  major challenges. One of them is conducting a subgroup level evaluation via graphical representation which is very detailed way to show an evaluation. another one is the significant fluctuation in pulse oximeter readings within the same individual over time, a phenomenon that appears to disproportionately affect Black patients, which is also demonstrated via detailed graphical representation.
This paper also shows comparison between RMSE values for different ML models across evaluation sets.
Like the reference section.

**Summary Of The Paper:**

This paper shows a machine learning approach to reduce measurement bias by incorporating objective, non-invasive skin pigmentation metrics alongside individual physiological parameters. Their proposed model  demonstrated improvement of  SpO2 accuracy, yielding corrected SpO2 values across  ranges of skin pigmentation levels.

**Weaknesses:**

N/A